# Medication incidents in primary care medicine: a prospective study in the Swiss Sentinel Surveillance Network (*Sentinella*)

Markus Gnädinger,[1] Dieter Conen,[2] Lilli Herzig,[3,4] Milo A Puhan,[5] Alfred Staehelin,[1,4] Marco Zoller,[1] Alessandro Ceschi[6,7]

► Prepublication history and additional material is available. To view, please visit the journal (http://dx.doi.org/ 10.1136/bmjopen-2016-013658)

[1]Institute of PrimaryCare, University of Zurich, Zurich, Switzerland
[2]Swiss Patient Safety, Zurich, Switzerland
[3]Institute of Family Medicine, University of Lausanne, Lausanne, Switzerland
[4]Sentinel Surveillance Network, Swiss Federal Office of Public Health, Bern, Switzerland
[5]Epidemiology,Biostatistics, and Prevention Institute, University of Zurich, Zurich, Switzerland
[6]Division of Science, National Poisons Centre, Tox Info Suisse, Associated Institute of the University of Zurich, Zurich, Switzerland
[7]Department of Clinical Pharmacology and Toxicology, University Hospital Zurich, Zurich, Switzerland

**Correspondence to**
Dr Markus Gnädinger;
markus.gnaedinger@hin.ch

## ABSTRACT

**Objectives** To describe the type, frequency, seasonal and regional distribution of medication incidents in primary care in Switzerland and to elucidate possible risk factors for medication incidents.

**Design** Prospective surveillance study.

**Setting** Swiss primary healthcare, Swiss Sentinel Surveillance Network.

**Participants** Patients with drug treatment who experienced any *erroneous* event related to the medication process and interfering with normal treatment course, as judged by their physician. The 180 physicians in the study were general practitioners or paediatricians participating in the Swiss Federal Sentinel reporting system in 2015.

**Outcomes** *Primary:* medication incidents; *secondary:* potential risk factors like age, gender, polymedication, morbidity, care-dependency, previous hospitalisation.

**Results** The mean rates of detected medication incidents were 2.07 per general practitioner per year (46.5 per 1 00 000 contacts) and 0.15 per paediatrician per year (2.8 per 1 00 000 contacts), respectively. The following factors were associated with medication incidents (OR, 95% CI): higher age 1.004 per year (1.001; 1.006), care by community nurse 1.458 (1.025; 2.073) and care by an institution 1.802 (1.399; 2.323), chronic conditions 1.052 (1.029; 1.075) per condition, medications 1.052 (1.030; 1.074) per medication, as well as Thurgau Morbidity Index for stage 4: 1.292 (1.004; 1.662), stage 5: 1.420 (1.078; 1.868) and stage 6: 1.680 (1.178; 2.396), respectively. Most cases were linked to an incorrect dosage for a given patient, while prescription of an erroneous medication was the second most common error.

**Conclusions** Medication incidents are common in adult primary care, whereas they rarely occur in paediatrics. Older and multimorbid patients are at a particularly high risk for medication incidents. Reasons for medication incidents are diverse but often seem to be linked to communication problems.

## INTRODUCTION

Patient safety is a major concern in healthcare systems worldwide. Although most safety research has been conducted in the inpatient setting,[1] evidence indicates that

### Strengths and limitations of this study

► This is the first prospective and systematic collection of incident data in primary care in Switzerland that is characterised by three linguistic regions and two drug distribution systems.
► It was conducted by experienced physicians and with high response rates.
► There is likely—as expected—bias from selective and under-reporting or non-detection of medication incidents.

medical errors and adverse events pose a serious threat for patients in the primary care setting as well, since most patients receive ambulatory care.[2–4] Information about the frequency and outcomes of safety incidents in primary care is required to identify risks or 'hot spots', to prioritise them and to take action as needed.[5] The aim of the project was to describe the type, incidence, seasonal and regional distribution of medication incidents in primary care in Switzerland and to elucidate risk factors for medication incidents.

## METHODS
### Study design

We conducted a prospective surveillance study among primary care patients during 2015 to identify cases of medication incidents.

### Study population

The study population was any person undergoing drug treatment in general internal or paediatric practises participating the *Sentinella* network. The latter covers a representative sample of patients in primary care for Switzerland.[5] Founded in 1986, it was mainly designed to survey transmissible diseases. Later, it also assessed other health problems of public interest. It generates daily to weekly current data and covers the entire

geographic and linguistic regions of our country. Children, the mentally handicapped or the elderly were also included, all of whom might be at increased risk for medication errors.

## Medication incidents

We defined medication incidents as any erroneous event (as defined by the physician) related to the medication process and interfering with normal treatment course (eg, administration of an erroneous medication). We did not include lack of treatment effect, adverse drug reactions or drug-drug or drug-disease interactions without detectable treatment error. Nor did we consider medication incidents if patients refused to have them reported to the *Sentinella* system.

## Data sources

The study physicians recorded the patient's year of birth and gender on their weekly reporting form. After a maximum of 4 weeks, they had to fill in a detailed incident questionnaire (see online suplementary Appendix A). It comprised their *Sentinella* number and the calendar week of notification. Concerning the patients, they reported the living situation, several supposed risk factors for an incident as well as the following variables: hospitalisation during the previous year, care-dependency, number of drugs used chronically, number of chronic conditions and the Thurgau Morbidity Index (TMI, a seven-step Likert scale to measure morbidity in outpatients), to be compared with a denominator analysis (below). We further received a detailed description of the incident and proposals to avoid future incidents.

We got the annual number of patient-to-physician contacts (PPC) per practise from the *Sentinella* administration, as well as morbidity data from a fortnight cross-sectional denominator analysis of all patients consulting a *Sentinella* practice during weeks 11 or 12 (Gnädinger M, Herzig L, Ceschi A *et al*. Chronic Conditions and Multimorbidity in the Swiss Primary Care Population - A Prospective Study in the Swiss Sentinel Surveillance Network. *Sentinella*. Submitted).

We received the anonymised list of participating physicians, their specialty as well as the community size (Swiss Federal Statistical Office) and the linguistic region from the *Sentinella* administration. Information on Swiss medication sales in 2015 by the Anatomical Therapeutic Chemical (ATC) Classification System (an international classification for pharmaceutical products) was derived from Interpharma Switzerland (see online suplementary Appendix B).

The questionnaires were completed either electronically or on a paper/pencil version, the former as online SurveyMonkey questionnaires, the latter as sealed envelopes sent from the *Sentinella* administration. We had three study questionnaires: a detailed incident questionnaire, an initial one and a final one. The detailed incident questionnaire collected information about the circumstances of the incident and about the patient's properties (see online Supplementary Appendix A). The initial questionnaire served to describe the physician's practices in terms of number of physicians, availability of electronic patient history, electronic drug-drug interaction control system and drug distribution system (see online Supplementary Appendix C). The final questionnaire investigated non-reporting and difficulties with coding the morbidity variables (see online Supplementary Appendix D).

We constructed a variable 'incident relevance' from the items 'disturbance' and 'endangering' of the patients; if any of them was graded with 'medium' or higher, the variable relevance was set to 'more', otherwise it was set to 'less'. Because of question ambiguity, we set coding of the item care-dependency to 'missing' for patients younger than 20 years of age. Evans' Index—a prognostic index—was calculated by simple addition of the number of chronic drug treatments with the number of chronic conditions.[6]

The following free text variables were manually coded: relationship of the incident to the suspected medication, preventability of the incident, reactions to the incident and proposals to avoid further incidents.

## Statistical methods

Values are given as frequencies, mean±SD or median (IQR) (as denoted with first quartile; forth quartile), depending on non-normal distribution or non-interval scaled data level. To assess the association of medication incidents with potential risk factors, we used the SPSS-GENLINMIXED procedure (a procedure that fits generalised linear mixed models). Clustering of patients was addressed by using a mixed binary logistic regression with the fixed factors of gender, year of birth, care-dependency, number of chronic drug treatments, number of chronic conditions and TMI as well as the physicians' practice number as a random factor; if one item was missing, the whole record was excluded from the analysis. Numerator was affiliation to the case group; denominator was all PPC during weeks 11 and 12. We used IBM SPSS V.23.

## RESULTS

### The Sentinella system

During the year 2015, 149 practices were enrolled to the *Sentinella* system. Of them, 144 practices were known to report regularly; their properties are listed in table 1. The *Sentinella* physicians are representative for the overall Swiss physician population.[5] Drugs were autodistributed by 42% of the study practices. Approximately half of the physicians had electronic and the other paper-based medical records. Systematic drug-drug interaction control systems were installed only by a minority (36.8%) of the physicians. During the year 2015 (which included unusually for calendar adaptation, 53 instead of 52 reporting weeks), the general practitioners (GPs) had 4456±2137 PPC; for the paediatricians (PEDs), these were 5297±2715 (mean±SD). In their initial study questionnaire, 25.0% of

| Table 1 | Characteristics of the reporting physicians in 2015 |
|---|---|
| **Physicians' gender** | |
| Male | 128 (71.1%) |
| Female | 52 (28.9%) |
| **Physicians' age class (years)** | |
| <40 | 12 (6.7%) |
| 40–49 | 44 (24.4%) |
| 50–59 | 66 (36.7%) |
| 60 and over | 58 (32.2%) |
| **Specialty** | |
| GP | 148 (82.2%) |
| PED | 32 (17.8%) |
| **Number of physicians in practice (n=144)** | |
| 1 | 69 (47.8%) |
| 2 | 39 (27.1%) |
| 3 | 17 (11.8%) |
| 4–5 | 8 (5.6%) |
| 6–9 | 7 (4.9%) |
| 10 and over | 4 (2.8%) |
| **Number of physicians per practice reporting to _Sentinella_ (n=144)** | |
| 1 | 119 (82.6%) |
| 2 | 19 (13.2%) |
| 3 | 5 (3.5%) |
| 8 | 1 (0.7%) |
| **Linguistic region** | |
| German | 122 (67.8%) |
| French | 44 (24.4%) |
| Italian | 14 (7.8%) |
| **Urbanity of the practice** | |
| Urban | 93 (51.7%) |
| Agglomeration | 60 (33.3%) |
| Rural | 27 (15.0%) |
| **Workload per week (hours)** | |
| <15 | 9 (5.0%) |
| 15–30 | 36 (20.0%) |
| >30 | 135 (75.0%) |
| **Drug distribution system** | |
| Dispensing by physician | 73 (42.2%) |
| Mixed system | 19 (10.6%) |
| Dispensing by pharmacy | 85 (47.2%) |
| **Electronic documentation** | |
| Yes | 89 (49.4%) |
| No | 91 (50.6%) |
| **Electronic interaction control** | |
| Yes | 65 (36.1%) |
| No | 115 (63.9%) |
| **Electronic prescription** | |
| Yes, with thesaurus | 62 (34.4%) |
| Yes, but without thesaurus | 24 (13.3%) |
| None | 94 (52.2%) |
| **Certification of the practice** | |
| Yes | 46 (25.6%) |
| None | 134 (74.4%) |
| | Continued |

| Table 1 | Continued |
|---|---|
| **Staff meetings** | |
| Yes, at least monthly | 69 (38.3%) |
| Yes, but less frequently | 70 (38.9%) |
| None | 41 (22.8%) |
| **Quality circle participation** | |
| Yes, at least monthly | 134 (74.4%) |
| Yes, but less frequently | 23 (12.8%) |
| None | 23 (12.8%) |

the reporting physicians admitted to be working part time (ie, <30 hours per week).

### Study flow

During the year 2015, we received 216 incident notifications (figure 1). In 11 cases, we did not receive the detailed notification form. Eight cases had to be removed from the database because they fulfilled the exclusion criterion (ie, 'adverse drug reactions _without_ detectable error'), and one case had to be removed because of double data entry. This led to 197 cases which could be analysed. The distribution of the monthly incident notifications throughout 2015 is depicted in online Supplementary Figure e1, Appendix E), and the distribution of the numbers of cases reported by each practice in figure 2.

### Description of patients

Table 2 lists age, gender and geographic distribution as well as the physician-to-patient relationship and the observer of the incident. Only three cases evolved in PED practices. No statistically significant differences were found between the two relevance classes of the incidents.

### Number of incidents, non-reporting

In the GPs 194 incidents, 148 physicians, 4456 yearly PPC led to 1.31 incidents per physician per year or 29.4 per 100 000 PPC; in PEDs 3 incidents, 32 physicians, 5297 yearly PPC led to 0.1 incident per physician per year or 1.8 per 100 000 PPC.

To evaluate the _non-reporting of incidents_, we asked the physicians in the final study questionnaire. Out of 180 actively reporting physicians, we received 145

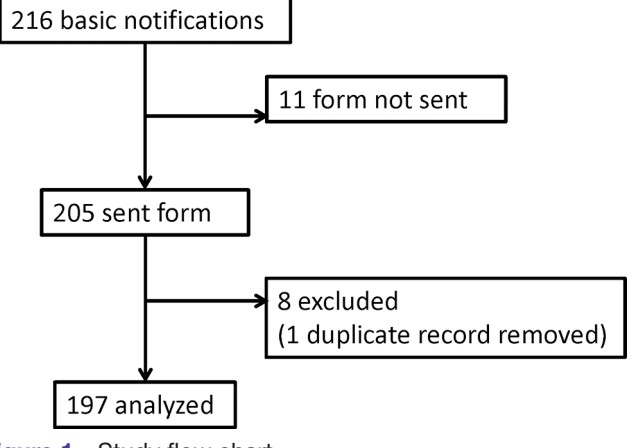

**Figure 1** Study flow chart.

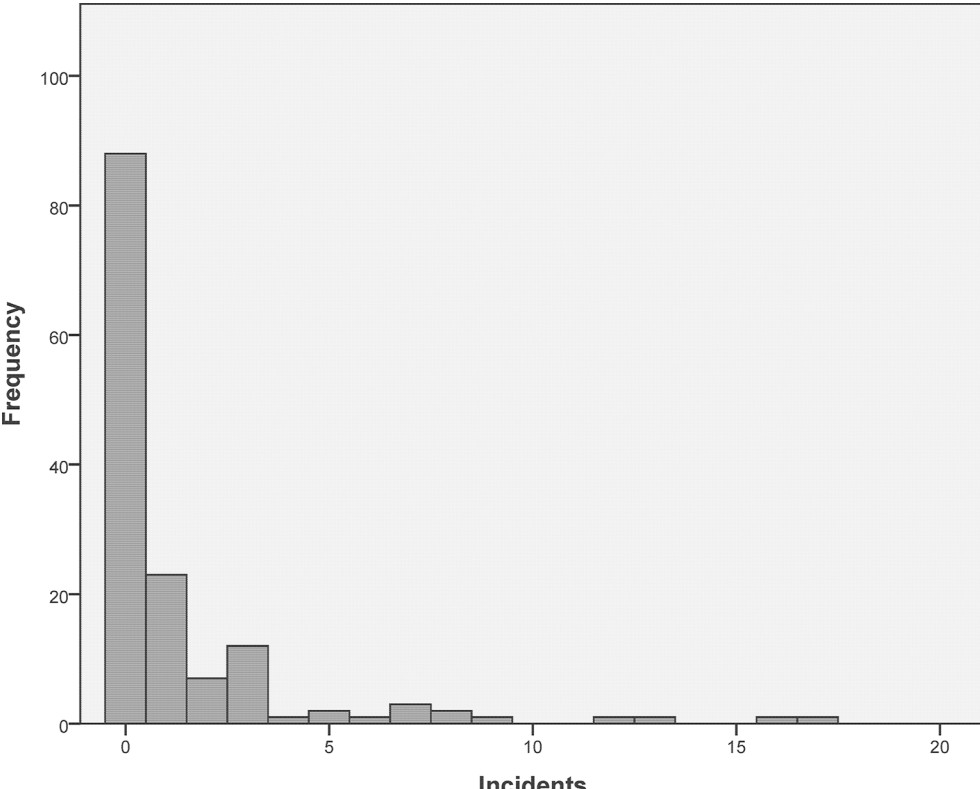

**Figure 2** Distribution of the number of cases reported by practice.

questionnaires (80.6% response rate). To our question: 'Did you not report medication incidents that you had noticed during the last year?', they answered: 'Never or almost' 110 (75.9%), 'yes, but seldom' 22 (15.2%), 'yes, frequently' 9 (6.2%), 'always, or almost' 4 (2.8%). Reasons for not reporting incidents were 'lack of time' or 'forgetfulness'. If we speculate that these answers represent reporting rates of 76%–100%, 51%–75%, 26%–50% or 0%–25%, respectively, we could divide the observed rates by the middle of the reporting classes: 0.875, 0.625, 0.375 and 0.125. By doing so, we calculated a rate of 50% of under-reporting; if we furthermore consider the 5% of the incidents where the questionnaires were not sent, this rate increases to 58%. We therefore have to multiply the observed rates with a factor of 1.58 resulting in the following rates of detected incidents: GP 2.07 per physician per year, 46.5 per 100 000 PPC and PED 0.15 per physician per year, 2.8 per 100 000 PPC.

### Types and causes of incidents, organ systems involved, preventability

The *types of error* are listed in online Supplementary table e1, Appendix E and figure 3; in 26 cases, more than one was mentioned. Most cases were linked to an incorrect dosage for a given patient, while prescription of an erroneous medication was the second most common error.

Most errors concerned orally ingested medication. Errors in application of *parenteral drugs* were reported less frequently; in our study, they comprised insulin (one case) and vaccinations (three cases). There were cases of an incorrect (influenza vs antitetanus) or incomplete

(Boostrix vs Boostrix-Polio) vaccination and of undue vaccination (a third anti-HPV vaccine). One case was due to a communication problem within the practice staff, another to a vaccination in absence of the vaccination card.

Three cases concerned paediatric patients, all were linked to inadequate dosing, two times of an antibiotic prescription (cotrimoxazole, co-amoxicillin) and one time of an antiemetic drug (thiethylperazine).

In 89 of the 195 cases, *organ system* damage was reported, in 13 cases more than one organ system was involved, most frequently the central nervous system (figure 4) (see also the online Supplementary table e2, Appendix E). Possible *triggers* of the incident were reported in 194 of the 197 cases; in 45 cases, there was more than one single reason, most frequently lacking alertness of the reporting physicians or their staff (see online Supplementary table e3, Appendix E). When asked, who might be the '*responsible*' for the incident, 191 replied. The most common person or institution possibly responsible for the medication incident was: the reporting physician 41 (21.5%), followed by the practice nurse 26 (13.6%), the institution where the patient lives 33 (17.3%), the pharmacy 7 (3.7%), the hospital 12 (6.3%), the community nurse 4 (2.1%), the patients or their proxies 15 (7.9%) and the manufacturer 2 (1.0%); in 9 cases (4.6%) this was unclear, in 37 cases (19.4%), there was more than source one to blame. *Preventability* of the incidents was classified (by our study board): unlikely in 6 cases (3.0%), possible in 58 (29.4%), probable in 114 (57.9%) and definite in 19 (9.6%).

**Table 2** General description of the cases

| | Relevance | | All |
| --- | --- | --- | --- |
| | **Less** | **More** | |
| Number of cases | 124 | 73 | 197 |
| Patient's age | 69.2±20.6 | 69.4±21.2 | 69.3±20.8 |
| Patient's gender, % males | 40.3 | 32.9 | 37.6 |
| Physician's specialty % paediatricians | 1.6 | 1.4 | 1.5 |
| Linguistic region, % | | | |
| German | 75.0 | 68.5 | 72.6 |
| French | 18.5 | 28.8 | 22.3 |
| Italian | 6.5 | 2.7 | 5.1 |
| Physician-to-patient relationship, % | | | |
| Own family physician | 86.3 | 78.1 | 83.2 |
| Urgency / holiday replacing | 0.8 | 4.1 | 2.0 |
| Institution physician | 11.3 | 17.8 | 13.7 |
| Other | 1.6 | 0.0 | 1.0 |
| Observer of the incident; % | | | |
| Physician/practisepractice staff | 50.0 | 50.7 | 50.3 |
| Patient / proxies | 21.8 | 23.3 | 22.3 |
| Community nurse | 1.6 | 4.1 | 2.5 |
| Institution (where patient lives) | 15.3 | 16.4 | 15.7 |
| Hospital | 0.8 | 1.4 | 1.0 |
| Other physicians | 2.4 | 0.0 | 1.5 |
| Pharmacist | 7.3 | 4.1 | 6.1 |
| Other | 0.8 | 0.0 | 0.5 |

### Endangering and disturbances

In 192 of the 197 cases, the item 'patients' *endangering*' as estimated by the physicians was answered. In 39 cases (20.3%), there was no endangering, in 83 (43.2%) light, in 51 (26.6%) moderate, in 19 (9.9%) severe. The answers to item '*disturbances* caused by the incident' are listed in table 3. Out of the 197 incidents, 74 (37.5%) were classified as 'more' relevant (see the 'Methods' section and table 2).

### Interface problems, orientation, predictability, repeat incidents

The presence of *interface problems* was reported in 64 of 197 cases (32.4%); these were with a hospital in 28 cases (43.8%), with an institution in 14 (21.9%), with a community nurse in 6 (9.4%), with a pharmacist in 9 (14.0%), with a specialist in 3 (4.7%) and with others in 4 (6.3%). In 184 cases, we received information about *informing* patients about the incident; in 98 cases, the patient was oriented by the reporting physician or his staff (50.3%), non-orientation was stated: because the patient was not able to understand (children, demented) in 26 cases (14.1%), because the problem had already been solved and the notification would have unduly disturbed the confidence in 18 cases (9.8%), because the patient or the proxies had observed it themselves in 23 cases (12.5%), because the patient had already been oriented by others in 7 cases (3.8%) or because of another reason in 12 cases (6.5%). As for *predictability* of the incident, we received 183 valid answers; of them 82 (44.8%) stated 'yes, in the given constellation, the incident was to be expected'. When asked whether they had already reported a *similar incident* in the study, 29 out of 196 (14.8%) answered in the affirmative.

### Risk factors to undergo a medication incident

To detect patient *risk factors* to undergo an incident, we compared the incident data with those of a fortnight denominator analysis; the following univariate factors accumulated preferentially in incident patients: higher age, care-dependency, higher numbers of chronic conditions or medications (or higher Evans' Index) as well as higher TMI (table 4). In the logistic regression, only inpatient care by an institution remained a significant factor. Other suspected risk factors are listed in online Supplementary table e4, Appendix E); these items were not included in the denominator study and therefore lack a comparator. Within the patient group with a more as compared with a less relevant incident, only psychiatric illness reached a significantly increased proportion. We did not detect major differences between the two drug distribution systems.

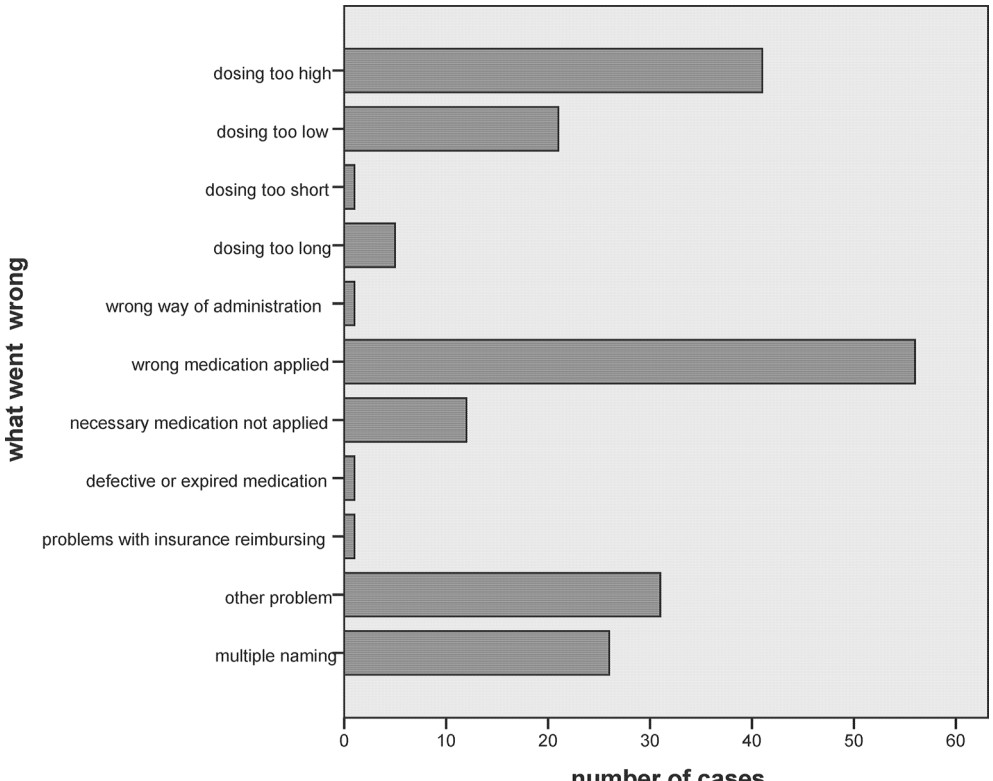

**Figure 3** Type of error.

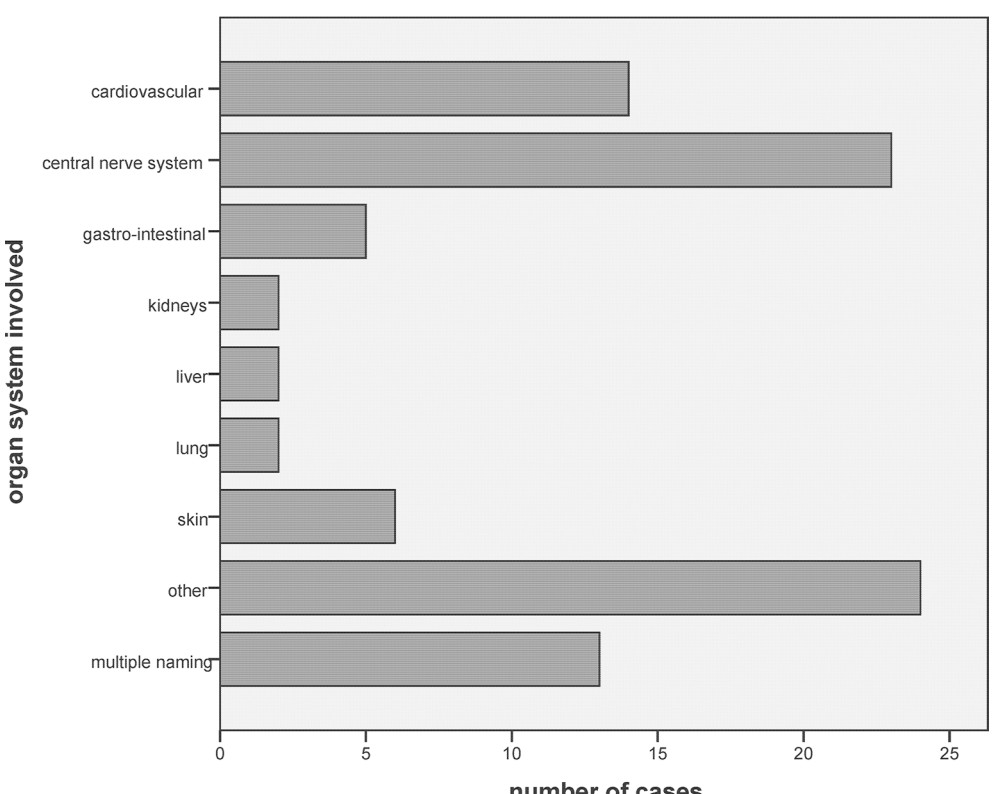

**Figure 4** Organ system involved.

**Table 3** Disturbances after the incident

| Item | N | Per cent |
|---|---|---|
| Severity of disturbance | | |
| No symptoms but pathological laboratory tests | 15 | 8.0 |
| Light | 44 | 23.4 |
| Moderate | 22 | 11.7 |
| Severe | 10 | 5.3 |
| Fatality | 0 | 0.0 |
| Subtotal (this is the base of the next rows) | 91 | 48.4 |
| No symptoms, normal (or no) laboratory tests | 97 | 51.6 |
| Total | 188 | 100.0 |
| Missing data | 9 | n.a. |
| All patients | 197 | n.a. |
| Time until recovery | | |
| Hours | 26 | 28.5 |
| Days | 41 | 45.1 |
| Weeks | 15 | 16.5 |
| Not yet known or missing information | 9 | 9.9 |
| All patients with disturbances | 91 | 100.0 |
| Recovering | | |
| Without sequels | 78 | 85.8 |
| With light-to-moderate sequels | 2 | 2.2 |
| With severe sequels or fatality* | 5 | 5.4 |
| Not yet known or missing information | 6 | 6.6 |
| All patients with disturbances | 91 | 100.0 |
| Treatment / surveillance | | |
| Not needed | 48 | 52.7 |
| Ambulatory care | 33 | 36.3 |
| Hospital care** | 7 | 7.7 |
| Missing information | 3 | 3.3 |
| All patients with disturbances | 91 | 100.0 |

\* In one case, there was a reduced kidney function, the other cases remained unclear, and no fatalities were reported.
\*\* Two cases had to be surveilled in the emergency room, the hospital stays were: intoxications with thiethylperazine, with fenoterol plus ipratropium bromide, with zolpidem, further a derailed diabetes type 2 (after missed treatment with metformin), and a gastrointestinal haemorrhage in a patient where antithrombotic treatment with rivaroxaban was not communicated to the physician.
n.a., not available.

### Relationship between incidents and drug class, ATC group

We assessed semi-quantitatively whether the class of the suspected drug was causally linked to the emergence of the incident; the frequencies were: none 49 (25.1%), unlikely 31 (15.9%), possible 77 (39.5%), probable 34 (17.4%), definite 4 (2.1%), did not apply 2 (not available). ATC codes of the suspected medications are listed in table 5. Most naming concerned the groups C 'cardiovascular' (23.2%) and N 'nervous system' (22.1%). Among the medication classes judged to be *related* to the incident (n=38), the most frequently named groups were oral anticoagulants: rivaroxaban 9, phenprocoumon 7 and acenocoumarol 2 cases; this explains the seven fold increased relative risk of ATC group B as compared with sales. As an example for errors *without* relation to the drug class, we must mention the 17 cases of institutionalised patients ingesting medications scheduled for other residents, in most cases this was a person eating at the same table, but in other cases the person in care mixed up patient names.

### Reactions to the incident and suggestions to prevent further ones

In 141 of the 197 cases, the physicians reported to have changed something after the incident, in 13 cases this was more than one *single action*, mostly often named were communication with other caregivers or better instruction of patients (see online Supplementary table e5, Appendix E). The respondents were asked for proposals how to *prevent* future incidents of the reported type; 125 of them made a total of 243 suggestions (see online Supplementary table e6, Appendix E). Of these, 37 (15.2%) were related to accurate medication lists, 10 (4.1%) to better patient instructions, 38 (15.6%) to organisation of regular follow-up controls and 55 (22.6%) involved organisational changes within the practice and its staff.

### DISCUSSION

In a representative group of primary care physicians,[7] we found approximately one case of a medication incident per GP and year.

### Incident rates

Incident rates in this study were similar to those found in other studies. We calculated the rates of detected medication incidents as follows: GP 2.07 per physician and year, 46.5 per 100 000 PPC and PED 0.15 per physician and year, 2.8 per 100 000 PPC. Medication incidents may make up a proportion of approximately one-third of all incidents[8]; the rates for *all safety incidents* may amount 6.20 per physician and year or 139.4 per 100 000 PPC in GPs and 0.45 per physician and year or 8.4 per 100 000 PPC in PEDs.

In Australian primary care patients, an incident rate of 4.98 per GP and year was reported; this is close to our estimated rate of 6.20; however, there was no subtyping of the incidents.[9] In a 3-year study, O'Beirne *et al* reported a rate of 1.8 safety incidents per year and physician.[8] In a literature review of western countries' publications, Sandars and Esmail described a rate of 5 to 80 errors per 100 000 consultations,[2] a rate somewhat lower than the 139.4 cases per 100 000 consultations estimated in our study. Kuo *et al* reported a proportion of 15% of all incidents to be related to medication[10]; since we estimated a proportion of 33% would give rise to even higher rates of all safety incidents when calculated from our study database. An

**Table 4** Possible risk factors for incident as compared with a denominator analysis during calendar weeks 11 and 12

| Item | Patient group | | OR (95% CI) (crude) | OR (95% CI) (adjusted) |
|---|---|---|---|---|
| | Denominator | Incidents | | |
| Number of observations | 26 852 | 197 | / | / |
| Age (mean±SD), years | 46.7±27.5 | 69.3±20.8 | 1.004 (1.001 to 1.006)§* | 1.001 (0.996 to 1.005)§ |
| Gender | | | | |
| Male | 47.0 | 37.6 | 1 | 1 |
| Female | 53.0 | 62.4 | 1.048 (0.916 to 1.197) | 1.055 (0.878 to 1.196) |
| Care-dependency, number (%)‡† | | | | |
| None | 16 335 (85.5%) | 96 (51.6%) | 1 | 1 |
| Yes, by proxies | 954 (5.0%) | 12 (6.5%) | 1.121 (0.789 to 1.594) | 0.979 (0.674 to 1.423) |
| Yes, by community nurse | 723 (3.8%) | 22 (11.8%) | 1.458 (1.025 to 2.073)* | 1.201 (0.821 to 1.758) |
| Yes, by institution | 1099 (5.8%) | 56 (30.1%) | 1.802 (1.399 to 2.323)*** | 1.528 (1.141 to 2.046)* |
| Number of conditions (median, IQR) | 2 (0 to 4) | 5 (3 to 7) | 1.052 (1.029 to 1.075)§*** | 1.030 (0.994 to 1.067)§ |
| Number of chronic active treatments (median, IQR) | 1 (0 to 4) | 6 (3 to 9) | 1.052 (1.030 to 1.074)§*** | 1.030 (0.995 to 1.067)§ |
| Evans' Index (median, IQR) | 3 (0 to 8) | 11 (6 to 17) | 1.009 (1.005 to 1.013)§*** | n.a.¶ |
| Thurgau Morbidity Index value (%)† | | | | |
| 0 | 8463 (31.5%) | 24 (12.2%) | 1 | 1 |
| 1 | 3611 (13.4%) | 8 (4.1%) | 0.989 (0.787 to 1.242) | 0.908 (0.694 to 1.190) |
| 2 | 4102 (15.3%) | 23 (11.7%) | 1.049 (0.847 to 1.300) | 0.898 (0.685 to 1.169) |
| 3 | 3877 (14.4%) | 39 (19.8%) | 1.131 (0.914 to 1.399) | 0.830 (0.611 to 1.127) |
| 4 | 2119 (7.9%) | 39 (19.8%) | 1.292 (1.004 to 1.662)* | 0.901 (0.643 to 1.265) |
| 5 | 1539 (5.7%) | 38 (19.3%) | 1.420 (1.078 to 1.868)** | 0.823 (0.547 to 1.239) |
| 6 | 709 (2.6%) | 26 (13.2%) | 1.680 (1.178 to 2.396)*** | 0.866 (0.523 to 1.436) |
| missing values | 18 | 0 | | |

*p<0.05, **p<0.01, ***p<0.001—significance levels.
†Because GENLINMIXED procedure was not able to process ordinal scaled variables, correlations between study group and TMI or care-dependency were tested with Spearman's rho: the correlation coefficients were +0.075 or +0.094, respectively; p<0.001.
‡Because of question ambiguity this analysis was restricted to adult patients (age >19 years); this led to 19 812 valid observations in the denominator and 183 in the incident groups.
§Per one conditions, medication, year or index point.
¶Because Evans' Index is a composite of condition and medication numbers, it was not included in the multiple regression analysis.
n.a, not available.

**Table 5    ACT groups of suspected medications**

| ATC class | | All incidents | Incidents with *probable* or *definite* relationship with medication | Relative risk (percentages left by right column) | Swiss 2015 sales* (number of packages) |
|---|---|---|---|---|---|
| A | Alimentary tract and metabolism | 28 (14.6%) | 6 (15.8%) | 1.06 | 31 455 252 (14.9%) |
| B | Blood and blood-forming organs | 23 (12.0%) | 7 (18.4%) | 7.08 | 5 507 624 (2.6%) |
| C | Cardiovascular system | 44 (22.9%) | 1 (2.6%) | 0.35 | 16 027 143 (7.5%) |
| D | Dermatologics | 1 (0.5%) | 0 (0.0%) | 0.00 | 17 314 810 (8.2%) |
| G | Genitourinary system and sex hormones | 1 (0.5%) | 0 (0.0%) | 0.00 | 7 936 641 (3.7%) |
| H | Systemic hormonal preparations (excluding sex hormones and insulins) | 7 (3.6%) | 1 (2.6%) | 2.00 | 2 875 760 (1.3%) |
| J | Anti-infectives for systemic use | 23 (12.0%) | 6 (15.8%) | 3.95 | 8 444 623 (4.0%) |
| K | Infusion liquids | 0 (0.0%) | 0 (0.0%) | 0.00 | 24 158 749 (11.5%) |
| L | Antineoplastic and immunomodulating agents | 5 (2.6%) | 1 (2.6%) | 2.89 | 1 934 950 (0.9%) |
| M | Musculoskeletal system | 4 (2.1%) | 2 (5.3%) | 0.76 | 14 787 413 (7.0%) |
| N | Nervous system | 43 (22.4%) | 10 (26.3%) | 1.30 | 42 690 195 (20.2%) |
| P | Antiparasitic products, insecticides and repellents | 1 (0.5%) | 0 (0.0%) | 0.00 | 462 559 (0.2%) |
| R | Respiratory system | 7 (3.6%) | 3 (7.9%) | 0.57 | 28 837 468 (13.7%) |
| S | Sensory organs | 2 (1.0%) | 0 (0.0%) | 0.00 | 7 658 311 (3.6%) |
| T | Diagnostic use | 0 (0.0%) | 0 (0.0%) | 0.00 | 43 184 (0.0%) |
| V | Various | 3 (1.6%) | 1 (2.6%) | 6.50 | 855 707 (0.4%) |
| | Total | 192 (100.0%) | 38 (100%) | 1.0 | 210 990 389 (100.0%) |
| | Does not apply | 5 | 0 | n.a. | n.a. |

\* Information by Interpharma Switzerland (see online Supplementary Appendix B).
Relative risk of drugs with probable or definite relationship with the incident as compared with sales proportions.
n.a., not available.

11% group of the physicians worked part time, the rate per full-time physician per year may therefore be somewhat higher than calculated. The 13-fold higher incident rate in GPs as compared with PEDs is not surprising given the lower medication rates in children. A recent British study confirmed this much lower rate of incidents in children; out of 46 902 family practice safety reports, 1788 concerned children (26-times less than adults).[11]

### Definition of incidents and reliability of reporting

On the other hand, incident rates may be influenced by their definition. Gandhi *et al* coined the term 'avoidable adverse drug reaction'.[12] In our study, we explicitly excluded adverse drug reactions *without* detectable error. Runciman *et al* defined a patient safety incident as follows: 'An event or a circumstance that could have resulted or did result in unnecessary harm to a patient'.[13] An intuitive definition of a medical error was given by Makeham *et al*: "That was a threat to patient wellbeing and should not happen. I don't want it to happen again".[14] As shown in online Supplementary figure e1, Appendix E), reporting frequency was higher at the beginning of the study as compared with the later course of it. This could reflect some loss of interest or forgetfulness by the reporting physicians. As calculated from our final questionnaire after the study, the reporting physicians failed to report about one in three cases of the detected medication errors. Non-detection of incidents may even be more frequent than non-reporting of observed incidents. A missed possible drug-drug interaction may be detected by chart review, a documentation error would probably have been found only in 1:1 supervision, which is very time-consuming, costly and may additionally influence performance of the observed physician. It is therefore virtually impossible to decipher the real rate of non-detected incidents. The problems in detection of incidents were the reason to postulate a 'mix of methods' as needed to identify adverse events in general practice.[15]

### Other approaches to investigate safety incidents

In contrast to our prospective investigation, there are other methods to approach safety incidents. In a retrospective, semi-quantitative analysis, Gehring *et al* investigated safety incidents in Swiss primary

care[16]; among 23 predefined classes of safety incidents, the respondents admitted 15 to have occurred at least yearly—four of them being linked to drug treatment. Another approach to incidents is to ask patients as performed by Mira et al[17]; they interviewed patients (>65 years with five or more chronic drug treatments) and found that 75% of the patients reported to have been affected by at least one medication error during the previous 12months. We did not collect data on appropriateness of treatment; admittedly medication inappropriate for some groups of patients (elderly or patients with impaired renal function) may provoke incidents.[18] A recent Scottish study demonstrated an impressive reduction in high-risk prescriptions (non-steroidal anti-inflammatory drugs, antiplatelet and anticoagulation) as well as hospital admissions for gastrointestinal ulcers or heart failure by a combination of educational measures, informatics and financial incentives.[19]

### Type, consequences, causes and preventability of error

Most cases in this study involved application of an erroneous dosing or of a wrong medication, although non-application of necessary medication was also frequent. The distribution of the incidents was similar to that reported in other studies.[10 16 17 20] When a wrong medication was dispensed, the error was often caused by confounding of prepared medication in home residents. The classic case of non-application of a necessary medication was the missed reuptake of anticoagulation after an operation. Non-application of necessary drugs seems to be a relevant source of unnecessary harm to patients.[21] The causing of the incidents was similar as reported by others.[10 16 17 20] Most studies found—as in our patients—lacking alertness of and communication problems within practice staff, but there was a large variety of other topics. All three paediatric cases were linked to prescription of an inadequate dosing for age and weight. More than half of the patients did not have any disturbances after the incident; otherwise in most cases, the nervous system was affected. No fatalities were reported, but seven patients (3.5%) needed stationary care. In 2004, Pirohamed et al published a study on adverse drug reactions as a cause for admission to hospital; they found that 6.5% of all hospitalisations and 4.0% of all hospital days were caused by adverse drug reactions, 72% of them being preventable and 2% leading to death[22]; however, this study included all cases, and therefore a majority of cases of adverse drug reactions *without* a detectable error. An older Swiss study was published by Livio et al; out of 3195 hospitalisations, she identified 229 cases (7.2%) as probably caused by adverse drug reactions.[23] In that study, 32% of the events were classified as being preventable (which does not strictly mean that an error had caused the incident), and in 6% of the cases, fatalities were reported. The results on outpatients contrast to inpatients where 1 in 10

drug administrations was described to be erroneous.[24] Medical errors seem to be the third leading cause of death in the USA.[25]

### Risk factors

We were able to identify some risk factors such as higher age, care-dependency, higher numbers of chronic conditions or medications (and higher Evans' Index) as well as higher TMI. However, adjusted in the logistic regression analysis, only care-dependency remained a significant risk factor, but this may be due to the small number of observations, which possibly precluded less important risk factors to be detected. The only significant risk factor for undergoing an incident of higher relevance was psychiatric disease. Several factors which reflect quite well the results of our study have been described in the literature to correlate with proneness to undergo a medication incident: mainly the number of drugs ingested,[12 26] but also higher or young age[27] and morbidity[28 29] have been described. Most incidents concerned ATC groups N 'nervous system' or C 'cardiovascular', which were also of the mostly sold drugs; an exception was group B with anticoagulants and a sevenfold increased relative risk as compared with Swiss sales in 2015. It seems wise to be alert to avoid errors when prescribing medication of these groups. The prevailing position of anticoagulants (18.5% of cases) was described also by Field et al.[29] Otherwise, the repartition of our suspected drugs was similar to other primary care studies,[10 29] a literature review[30] or a theoretical paper.[20]

### Limitations

There is likely as expected bias from selective and under-reporting or non-detection of medication incidents. Therefore, the true rate of incidents could only be estimated, and the proportions of incident characteristics would probably substantially have been altered by a more complete recording of incidents.

### CONCLUSION

Medication incidents are common in general medicine, whereas they rarely occur in paediatrics, in which polypharmacy is less prevalent. Reasons for medication incidents are diverse but often seem to be linked to communication problems. Older and multimorbid patients are at a particularly high risk for medication incidents.

**Acknowledgements** We are grateful to Lee Wennerberg for the English language corrections, Dr Sven Staender, Männedorf for the helpful comments, and Simon Gnädinger, Zurich for the manual coding of free text items. We thank the *Sentinella* programme commission for their support, the reporting physicians of *Sentinella* for their unflagging enthusiastic collection of data, and the Federal Office of Public Health for providing data and translating the questionnaires into French.

**Contributors** MG led the study, did the pilot study (questionnaire development, data entering and processing), wrote all documents, did all the contacts with the *Sentinella* administration, ethics committee and others, programmed the electronic questionnaires, entered hand-written questionnaires into the database, did the data processing and wrote the publication after data collection. AC is an expert on clinical pharmacology and drug safety. DC is an expert on patient safety. LH is French-speaking and helped to interpret the French questionnaires.

She is an expert on multimorbidity. She is a member of the *Sentinella* programme commission. MP is head of Epidemiology, Biostatistics & Prevention Institute. He is responsible for the sound methodology. AS had the idea for the study. He is vice president of the *Sentinella* programme commission. MZ is expert on electronic data exchange in primary care. All of them have seen all the study documents and have contributed intellectually to their elaboration. All have contributed to revise the draft of this publication and approve the submitted version of this publication.

**Funding** The study was funded by Bangerter-Rhyner Foundation, Basel.

**Competing interests** None declared.

**Ethics approval** Ethics commission of Canton Zurich Switzerland.

**Provenance and peer review** Not commissioned; externally peer reviewed.

**Data sharing statement** No additional data are available. Anonymized raw data can be demanded from the corresponding author.

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
