## [Reviewer comments · BMJ Open]

ARTICLE DETAILS

TITLE (PROVISIONAL)	Medication Incidents in Primary Care Medicine: a Prospective Study in the Swiss Sentinel Surveillance Network (Sentinella)
AUTHORS	Gnädinger, Markus; Conen, Dieter; Herzig, Lilli; Puhan, Milo; Staehelin, Alfred; Zoller, Marco; Ceschi, Alessandro

VERSION 1 - REVIEW

REVIEWER	Meredith Makeham Australian Institute of Health Innovation Macquarie University Sydney Australia
REVIEW RETURNED	29-Aug-2016

GENERAL COMMENTS	1. More clarity please around the Results sentence in the abstract that starts "The following factors were associated with medication incidents...". For example, "care dependency 1.456 (1.025;2.073) for care by community nurse" doesn't have a clear meaning, and there is variability with the positioning of the 95% CIs in the sentences.2. Suggest you should consider an additional reference for discussion of proportion of reported incidents that related to medications. Authors used an estimate from a Canadian study with quite different methodology, but look at Makeham et al "Patient safety events reported in general practice: a taxonomy" in Qual Safety Health Care 2008 which reported the breakdown of the TAPS study error reports using a categorisation showing medication versus other incidents. It was slightly over 30%.3. There are some confusing sections in the results. Perhaps some of the choices for English expression could be reworded - eg the use of the term "incidents per physician and year" might be better as "per physician per year"? Also "Most errors concerned orally applied medication". Do you mean orally ingested? Also the use of "applied" with parenteral drugs. Applied implies topical application. Also "When discerning patients with a more from them with a less relevant incident" should be reworded. A percentage is missing in the ATC group section after '2'.4. I feel the discussion needs a little English language finessing, in particular 'Type, consequences, causes and preventability of error' section. Eg "The prototype of wrong medication was confounding of prepared medication in home residents" and "The repartition of the incidents was similar as reported by others". The term "sensu strictu". "Concerning circumstances, most naming was - as found in our patients - lacking alertness and communication problems within practice staff"... Also in this section, the last 2 sentences don't logically follow the discussion of primary care incidents.5. The use in various areas of the discussion of the term "hence" doesn't always seem to fit.
---

	6. In the Risk Factors section, "propensity factors" doesn't have a clear meaning to me. 7. I think the Prospects section sits quite uncomfortably with the rest of the paper. 8. Limitations are not discussed in any detail 9. There is little reference to any paediatric medication incident work to compare these results.
--	---

REVIEWER	Sara Garfield Imperial College Healthcare NHS Trust, UK
REVIEW RETURNED	30-Aug-2016

GENERAL COMMENTS	The reviewer provided a marked copy with additional comments. Please contact the publisher for full details.
--

REVIEWER	Stephen Senn Luxembourg Institute of Health Luxembourg None. However, I am Swiss so may have a prejudice in favour of work from Switzerland.
REVIEW RETURNED	19-Sep-2016

GENERAL COMMENTS	The presentation of your statistics is confusing or incomplete in places 1) Figure 2 is inappropriate. You have suppressed the zero class. Please see attached for an alternative   <caption>Data for Figure 2: Number of physicians vs Incidents</caption>   Incidents Number of physicians    0125 122 210 312 42 52 62 73 82 91 101 111 121 131 141 151  	Incidents	Number of physicians	0	125	1	22	2	10	3	12	4	2	5	2	6	2	7	3	8	2	9	1	10	1	11	1	12	1	13	1	14	1	15	1
Incidents	Number of physicians																																		
0	125																																		
1	22																																		
2	10																																		
3	12																																		
4	2																																		
5	2																																		
6	2																																		
7	3																																		
8	2																																		
9	1																																		
10	1																																		
11	1																																		
12	1																																		
13	1																																		
14	1																																		
15	1																																		

2) For your logistic regression you need to make clear what the numerator is
 3) You state 'values are given as frequencies, mean \pm SD or median [interquartile range (IQR)]' but leave it up to the reader to guess which you are using when. Where you state

"the general practitioners (GPs) had 4,456 \pm 2,137 PPC; for the pediatricians (PEDs), these were 5,297 \pm 2,715."

I assume you are using median and IQR

4) The editors should note that without access to the original data the checks I have been able to make are essentially limited to the simple summary statistics below. The rest is taken on trust.

Check analysis of data in Gnädinger et al

Total physicians	Total incidents	Incidents per physician
180	197	1.09

Distribution of number of incidents by physician

Tally of Incidents by physician

Value	Frequency	Percentage	Cumulative	Cumulative %
0	124	68.9	124	68.9
1	23	12.8	147	81.7
2	7	3.9	154	85.6
3	12	6.7	166	92.2
4	1	0.6	167	92.8
5	2	1.1	169	93.9
6	1	0.6	170	94.4
7	3	1.7	173	96.1
8	2	1.1	175	97.2
9	1	0.6	176	97.8
12	1	0.6	177	98.3
13	1	0.6	178	98.9
16	1	0.6	179	99.4
17	1	0.6	180	100.0

Summary statistics for Incidents by physician

Number of observations = 180
 Number of missing values = 0
 Mean = 1.094
 Median = 0
 Minimum = 0
 Maximum = 17
 Sum of values = 197

Summary statistics for Physicians

Number of observations = 18

	Number of missing values = 0 Mean = 10 Median = 1 Minimum = 0 Maximum = 124 Sum of values = 180
--	--

VERSION 1 – AUTHOR RESPONSE

Answers to the comments of the reviewers

Reviewer	Comments	Answers
1	More clarity please around the Results sentence in the abstract that starts "The following factors were associated with medication incidents...". For example, "care dependency 1.456 (1.025;2.073) for care by community nurse" doesn't have a clear meaning, and there is variability with the positioning of the 95% CIs in the sentences.	Abstract (results): The parenthesis denoting the interquartile range was set immediately after the number denoting the median value. The wording was changed.
1	Suggest you should consider an additional reference for discussion of proportion of reported incidents that related to medications. Authors used an estimate from a Canadian study with quite different methodology, but look at Makeham et al "Patient safety events reported in general practice: a taxonomy" in Qual Safety Health Care 2008 which reported the breakdown of the TAPS study error reports using a categorization showing medication versus other incidents. It was slightly over 30%.	Discussion (incident rates): We changed reference 9 accordingly.
1	There are some confusing sections in the results. Perhaps some of the choices for English expression could be reworded - e.g. the use of the term "incidents per physician and year" might be better as "per physician per year"? Also "Most errors concerned orally applied medication". Do you mean orally ingested? Also, the use of "applied" with parenteral drugs. Applied implies topical application. Also "When discerning patients with a more from them with a less relevant incident" should be	Results (number of incidents): The corrections were done. Results (risk factors...): The wording was changed. Results (relationship...) We considered the two cases where the

	reworded. A percentage is missing in the ATC group section after '2'.	ATC group was classified to “does not apply” as missing and did not count them to the percentages.
1	I feel the discussion needs a little English language finessing, in particular 'Type, consequences, causes and preventability of error' section. E.g. "The prototype of wrong medication was confounding of prepared medication in home residents" and "The repartition of the incidents was similar as reported by others". The term "sensu strictu". "Concerning circumstances, most naming was - as found in our patients - lacking alertness and communication problems within practice staff"... Also in this section, the last 2 sentences don't logically follow the discussion of primary care incidents.	Discussion (Type, consequences...): The wording was changed. The two sentences concerning inpatients are now better introduced.
1	The use in various areas of the discussion of the term "hence" doesn't always seem to fit.	Discussion (various sections): We changed the wording.
1	In the Risk Factors section, "propensity factors" doesn't have a clear meaning to me.	Discussion (risk factors): We changed the wording.
1	I think the Prospects section sits quite uncomfortably with the rest of the paper.	Discussion (last paragraph): This section was deleted.
1	Limitations are not discussed in any detail	Discussion (new section “Limitations”): We included a new paragraph.
1	There is little reference to any pediatric medication incident work to compare these results.	Results (number of incidents) and discussion (type, consequences...): We included information on the three cases in pediatrics.
3	Figure 2 is inappropriate. You have suppressed the zero class. Please see attached for an alternative	Figure 2: Thank you for this important notion! However, in our study we have only information about practices and not about single physicians; the denominator is therefore 144 practices and not 180 physicians. We changed the figure 2 by including a “zero” incident category.
3	For your logistic regression, you need to make clear what the numerator is	Methods (statistical methods): We changed the methods section accordingly.

3	You state 'values are given as frequencies, mean \pm SD or median [interquartile range (IQR)]' but leave it up to the reader to guess which you are using when. Where you state "the general practitioners (GPs) had 4,456 \pm 2,137 PPC; for the pediatricians (PEDs), these were 5,297 \pm 2,715." I assume you are using median and IQR	Methods (statistical methods) and results (the Sentinella-system): We changed the methods section accordingly and added the meaning of the numbers to the mentioned paragraph.
3	The editors should note that without access to the original data the checks I have been able to make are essentially limited to the simple summary statistics attached. The rest is taken on trust.	On demand, we will provide an anonymized data set with pleasure for redoing our analyses by the reviewer.

VERSION 2 – REVIEW

REVIEWER	Sara Garfield Imperial College Healthcare NHS Trust
REVIEW RETURNED	25-Nov-2016

GENERAL COMMENTS	My comments on the first submission have not been addressed. The second submission is an improvement on the first but there are still issues which need to be addressed before publication (see my first review)
---

REVIEWER	Stephen Senn Luxembourg Institute of Health Luxembourg
REVIEW RETURNED	23-Nov-2016

GENERAL COMMENTS	I have only one minor point. It is an increasingly common misnomer in the public health literature to refer to models with multiple predictors as multivariate. As far as statistician are concerned, however, models with multiple right hand terms are not per se multivariate. It is models with multiple left hand terms that are multivariate. So, for example, predicting the joint response of diastolic and systolic blood pressure as a function of age is multivariate. Predicting DBP from age, sex, BMI, weekly exercise etc is not, even though all these many terms are in the model. From this perspective your use of multivariate, although regrettably common, is false. See, for example, https://www.ncbi.nlm.nih.gov/pmc/articles/PMC3518362/ Sometimes the term multivariable is used instead.
--

VERSION 2 – AUTHOR RESPONSE

Answers to the comments of the reviewers

We thank the reviewers for their helpful comments. These are listed below.

Reviewers number and initials	Text position	Comment	Answer
2. SG	Introduction p1	It is not sufficient to just reference the protocol. You need a justification for your study here.	We added a justification for our study, which is in accordance with our study protocol published in this journal.
idem	Methods / data sources	Define Thurgau-Morbidity-Index (TMI)	We added a brief definition as required. A detailed description is provided by reference 6.
idem	idem	Define ATC classes	We added the definition.
idem	idem	Refer to the detailed incident questionnaire.	We commented the detailed incident questionnaire.
idem	idem	The paragraph describing “incident relevance” is unclear.	We declared that this was a constructed variable.
idem	Methods / statistical methods	Define GENLINMIXED procedure.	We defined briefly the procedure as requested. A more detailed description of the procedure would not add clarity to the text in our

			opinion.
idem	Results / Number of incidents, non-reporting	The explanation about non-reporting variable should be in Methods section.	We mentioned this in the fourth paragraph of «data sources» in methods section.
idem	Results / types and causes...	(about parenteral drugs) denominator is per GP not per drug so not proportional? Maybe less parental drugs were prescribed. No numbers included above.	We changed the paragraph accordingly.
idem	Results / Endangering and disturbances	This section is unclear	We changed the paragraph and hope it is now clearer.
idem	Results / Interface problems	better “informing” than orienting	Thank you!
idem	Results / Reactions	better “action” than reaction	Thank you!
idem	Discussion	The discussion section is currently hard to follow. Perhaps it would help to draw out the key message for the reader at the beginning of each paragraph e.g. 'incident rates in this study were similar to those found in other studies ...	Thank you for this valuable suggestion. We have reformulated the paragraphs accordingly trying to draw out the key messages for the reader at the beginning of each paragraph.
idem	Discussion / Definition of incidents	Need to have a limitations section.	We absolutely agree with the reviewer and therefore added a limitations section (which

			was already included in the previous version).
idem	Discussion / Other approaches	Can you make this heading more informative?	We have reformulated to make the heading more informative.
idem	Conclusion	However less paediatric meds prescribed. Not necessarily lower rates of pediatric medication error.	Thank you for the comment. We have included this important point.
3. SS	Results and discussion	I have only one minor point. It is an increasingly common misnomer in the public health literature to refer to models with multiple predictors as multivariate. As far as statisticians are concerned, however, models with multiple right hand terms are not per se multivariate. It is models with multiple left hand terms that are multivariate. So, for example, predicting the joint response of diastolic and systolic blood pressure as a function of age is multivariate. Predicting DBP from age, sex, BMI, weekly exercise etc. is not, even though all these many terms are in the model. From this perspective, your use of multivariate, although regrettably common, is false. See, for example, https://www.ncbi.nlm.nih.gov/pmc/articles/PMC3518362/ Sometimes the term multivariable is used instead.	We thank the reviewer for the explanation and the clarification and changed therefore the term to "logistic regression".